# Selective ligand removal to improve accessibility of active sites in hierarchical MOFs for heterogeneous photocatalysis

Shaghayegh Naghdi [1], Alexey Cherevan [1], Ariane Giesriegl[1], Rémy Guillet-Nicolas[2,3], Santu Biswas[4], Tushar Gupta [1], Jia Wang[1], Thomas Haunold[1], Bernhard Christian Bayer [1], Günther Rupprechter [1], Maytal Caspary Toroker [4,5], Freddy Kleitz [2] & Dominik Eder [1✉]

Metal-organic frameworks (MOFs) are commended as photocatalysts for $H_2$ evolution and $CO_2$ reduction as they combine light-harvesting and catalytic functions with excellent reactant adsorption capabilities. For dynamic processes in liquid phase, the accessibility of active sites becomes a critical parameter as reactant diffusion is limited by the inherently small micropores. Our strategy is to introduce additional mesopores by selectively removing one ligand in mixed-ligand MOFs via thermolysis. Here we report photoactive MOFs of the MIL-125-Ti family with two distinct mesopore architectures resembling either large cavities or branching fractures. The ligand removal is highly selective and follows a 2-step process tunable by temperature and time. The introduction of mesopores and the associated formation of new active sites have improved the HER rates of the MOFs by up to 500%. We envision that this strategy will allow the purposeful engineering of hierarchical MOFs and advance their applicability in environmental and energy technologies.

---

[1] Institute of Material Chemistry, Technische Universität Wien, 1060 Vienna, Austria. [2] Department of Inorganic Chemistry - Functional Materials, Faculty of Chemistry, Universität Wien, 1090 Vienna, Austria. [3] Normandie University, ENSICAEN, UNICAEN, CNRS, Laboratoire Catalyse et Spectrochimie, 14050 Caen, France. [4] Department of Materials Science and Engineering, Technion - Israel Institute of Technology, Haifa 3600003, Israel. [5] The Nancy and Stephen Grand Technion Energy Program, Technion - Israel Institute of Technology, Haifa 3600003, Israel. ✉email: dominik.eder@tuwien.ac.at

Metal-organic frameworks (MOFs) have recently gained tremendous interest as an innovative class of functional materials owing to their catalytically active metal-oxo clusters (secondary building units, SBUs), functional organic linkers, easily tunable physicochemical properties, and large surface areas along with a well-ordered porous structure[1,2]. First experiments have also shown great potential for MOFs as photocatalysts[3–7], as they can combine tunable light-harvesting (e.g., through the organic linker) and catalytic functions (e.g., through the inorganic nodes) along with an excellent reactant adsorption capability. The proximity of catalytic, electronic, and adsorption sites of MOFs allows for high turnovers and selectivity. Moreover, the inorganic nodes of MOFs resemble molecular catalytic centers emulating those of homogenous catalysts. This enables heterogeneous frameworks with well-defined active sites that allow in-depth fundamental studies on the reaction kinetics and pathways[8]. Therefore, MOFs can be seen as model systems for exploring advanced concepts, such as bridging homogeneous and heterogeneous photocatalysis as well as single-site photocatalysis[9].

However, kinetic restrictions upon reactant diffusion through (small) micropores in MOFs impose a serious challenge for their effective use as (photo)catalysts, in particular in liquid reaction media[3,4,10,11]. A number of strategies have aimed at enhancing pore accessibility by expanding porosity in the original framework[12,13]. This can be achieved through synthesis or via post-synthetic modification that often relies on the reversible cleavage of selected bonds within the framework resulting in concurrent creation of larger pores[14]. The first category includes the use of elongated ligands[15,16], metal-ligand-fragment co-assembly[17,18], and imperfect crystallization[19,20] methods. The second category comprises the use of sacrificial soft or hard templates[21–23], modulator-induced defect formation[24], cation valence modulation[25], and ligand exchange[26]. All of these methods, however, suffer from their limited applicability (i.e., only a few MOF structures or ligand types are suitable), a lack of control over shape, location, and spatial distribution of the introduced porosity, and an undesired loss of crystallinity, often accompanied by considerable pore collapse upon synthesis or post-processing. Recently, Feng et al.[27] reported an intriguing route towards introducing hierarchical porosity in MOFs. This process, primarily demonstrated for UiO-66, involves the synthesis of mixed-ligand MOFs consisting of organic linkers with different thermal stability, followed by selective linker removal through thermolysis.

One can envision that such MOFs with dual-porosity will advance their use in various applications, particularly when operating in liquid media. These may include drug delivery/release, water purification, separation technologies, and liquid-phase catalysis[27,28]. We envision the greatest benefits for energy conversion technologies, involving electrocatalysis and photocatalysis, where - in addition to the enhanced porosity - the partial ligand removal is expected to create unsaturated metal sites that can serve as adsorption sites for reactants and co-catalysts and thus as potential catalytic centers.

In this work, we synthesize a series of photoactive MOFs with two distinct dual-porosity characteristics. We unravel the mechanistic steps of ligand removal through powerful in situ techniques coupled with density-functional theory (DFT) simulations and provide an in-depth analysis of the resulting hierarchical pore structures. Furthermore, we explore the impact of the type and size of the mesopores on the MOFs photocatalytic performance towards hydrogen production. Our results demonstrate that small variations of the synthesis conditions can introduce two distinct, structurally different mesoporous geometries without noticeably altering the overall framework

structure: cavities with ink-bottle pores and extended, interconnected fracture-type pores. Both geometries have significantly improved the hydrogen evolution rates of the MOFs, but it is the fracture-type geometry that has enhanced the activity by the highest factor. Therefore, this work provides an intriguing example of the potential of rationalized pore engineering towards advancing the performance of MOFs in various liquid phase applications.

## Results

**As-prepared mixed-ligand MOFs**. We synthesized a series of isostructural mixed-ligand MIL-125-Ti-based MOFs with various ratios of terephthalic acid (BDC) to amino-terephthalic acid (BDC-NH$_2$) (Supplementary Table 1), denominated as $x$NH$_2$-MIL-125-Ti, with $x$ being the percentage of BDC-NH$_2$. Due to the restrictions imposed by the limited solubility of the ligands, the samples were prepared following two routes that differ in the sequence of the precursor addition (Supplementary Fig. 1). For the samples with $2 \leq x \leq 10$, BDC-NH$_2$ was added to a solution of BDC and titanium (IV) isopropoxide (TTIP) (route 1), while for the samples with $50 \leq x \leq 80$ BDC was added to a solution of BDC-NH$_2$ and TTIP (route 2). For details refer to the Supplementary Information. The actual content of each ligand was quantified by $^1$HNMR spectroscopy (Supplementary Table 4) and fits very well with the targeted ratios.

X-ray diffraction (XRD) revealed that all synthesized MOFs are highly crystalline and consist of the expected crystal structure of MIL made of Ti$_8$O$_4$(OH)$_4$ clusters (SBU) connected via organic ligands (Supplementary Fig. 2a)[29,30]. It also confirms the absence of any impurity compounds or secondary phases. Note that the variation of the ligand has not affected the unit cell of the MOFs in a significant way, which demonstrates that both ligands are akin in terms of coordination directionality and size[31]. Interestingly, all samples exhibit a small diffraction peak at $2\theta = 13.6°$, whose intensity decreases with increasing BDC-NH$_2$ content in relation to the MOF peaks. This feature has not been reported before in literature. The simulated XRD pattern (Supplementary Fig. 2b) reveals that this peak is associated with the presence of solvent molecules (i.e., DMF) incorporated within the micropores during the synthesis. The peak intensity correlates inversely to the amount of the solvent, hence the peak increases with the evaporation of solvent molecules upon heat treatment (Supplementary Fig. 5).

The crystalline nature of the as-prepared mixed-ligand MOFs was further investigated by transmission electron microscopy (TEM). The insets in Fig. 1 show typical bright-field (BF) TEM images of individual particles of (a) 5%NH$_2$-MIL and (b) 50%

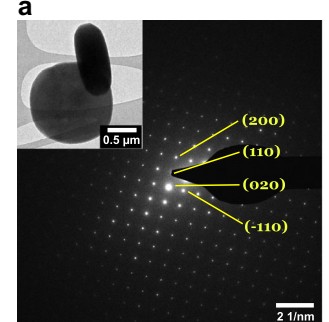
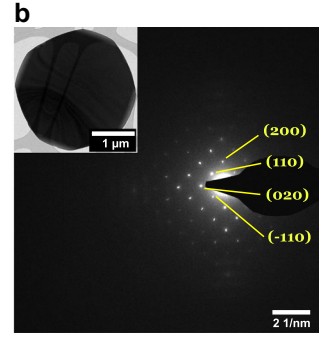

**Fig. 1 Crystallinity of as-prepared mixed-ligand MOFs.** SAED patterns recorded from an individual particle (BF TEM shown as insets) of as-prepared 5% (**a**) and 50%NH$_2$-MIL (**b**). All SAED patterns were indexed using the cif file of pure MIL-125-Ti-based MOF[30] viewed along the [001] zone axis with salient reflections labeled.

NH₂-MIL and reveal that the MOF particles are generally disk-shaped with regular facets and uniform size and shape. The observed homogenous contrast across individual particles suggests a single-crystalline nature. This was confirmed by the corresponding selected area electron diffraction (SAED) patterns, which confirm the expected structure and the absence of additional phases and impurities[30].

The formation of mixed-ligand MOFs was also analyzed by diffuse reflectance infrared fourier transform spectroscopy (DRIFTS). Supplementary Figure 3 shows the corresponding spectra of both single-ligand MOFs (MIL, NH₂-MIL) and two mixed-ligand MOFs (10% and 50%NH₂-MIL) at room temperature, highlighting the characteristic bands. The bands in the range between 3680 and 3660 cm⁻¹ can be assigned to intrinsic OH groups of the $Ti_8O_8(OH)_4$ clusters[32,33], as also reported for structurally closely-related Zr MOF[34]. Note that this feature is often a superposition of several closely positioned bands resulting from local structural variations caused by missing or singly coordinated ligands, trapped solvent molecules, or differently coordinated aqua and hydroxo groups[33].

The two broad bands centered at 3517 and 3384 cm⁻¹ and their pronounced shoulders at lower wavenumbers can be assigned to asymmetric and symmetric NH vibrations and thus confirm the presence of the BDC-NH₂ ligands in the respective MOFs. Note that in the mixed-ligand MOF, both bands are red-shifted compared to the single-ligand NH₂-MIL-125, which indicates a certain degree of H-bonding[35].

The bands at 1962 and 1935 cm⁻¹ feature the coordination of the ligands to the SBU characteristic for MIL (BDC) and NH₂-MIL (BDC-NH₂), respectively. Note that the mixed-ligand MOFs contain both bands with intensities that correspond well to the respective ligand ratios. A similar superposition of BDC-NH₂ and BDC is shown by the CH bending overtones in the range between 1800 and 1900 cm⁻¹. In the low wavenumber region, the pronounced CH band at 1020 cm⁻¹ seems unique for the BDC-containing MOFs and can thus be considered a fingerprint vibration. The peaks at 1160 and ~1710 cm⁻¹ belong to free C–O and C=O stretching vibrations, respectively, likely corresponding to the presence of dangling ligands[36]. The characteristic bands are summarized in Supplementary Table 2.

**Structural changes upon heat treatment**. The single-ligand and mixed-ligand MOFs were subjected to heating in the air up to 600 °C. Thermogravimetric analysis (TGA) reveals that both single-ligand MOFs experience a significant weight loss above 300 °C (Supplementary Fig. 4a, b), which is associated with an endothermic decomposition of the organic linkers (Supplementary Fig. 4c). Note, that the onset temperature of this weight loss depends on the type of ligand. The decomposition of BDC-NH₂ in NH₂-MIL proceeds via a 2-step process that starts just above 300 °C and concludes at 550 °C, the reason of which will be explained below. In contrast, the BDC linker in MIL decomposes within a narrower temperature window between 430 °C and 490 °C. The thermal removal of the organic compounds in the mixed-ligand MOFs follows this trend (Supplementary Fig. 4b, c), with the onset temperature increasing upon increasing amino content. This implies that the thermal removal of BDC-NH₂ in the amino-rich samples is kinetically more challenging than in the samples with lower BDC-NH₂ contents. Overall, these results confirm that BDC-NH₂ is more thermolabile than BDC, and that heating at temperatures between 300 °C and 350 °C allows for the selective removal of BDC-NH₂ from all MOF samples.

The structural changes upon heating up to 550 °C in the air were also investigated with in situ XRD (Supplementary Fig. 5a). Selected XRD patterns for the most characteristic temperatures

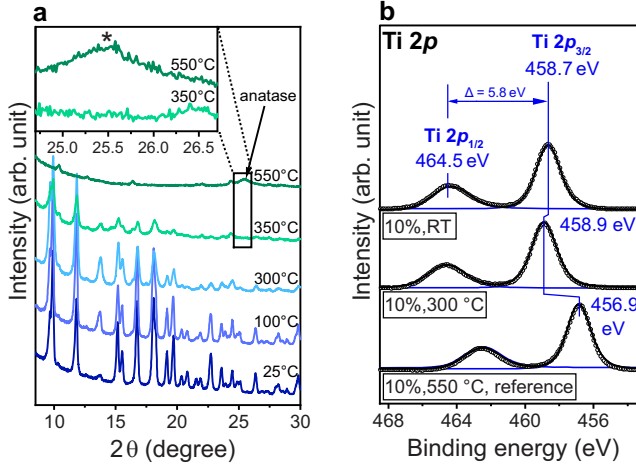

**Fig. 2 Characterization of 10%NH₂-MIL upon heat treatment. a** In situ XRD patterns collected during heating in air (0.5 mL min⁻¹) up to 550 °C (5 °C min⁻¹). **b** XPS Ti 2p spectra of as-prepared 10%NH₂-MIL, and after heat treatments in air at 300 °C and 550 °C (5 h). The shift in binding energies at 550 °C indicates the transformation of the MOF structure into TiO₂, which correlates well with the appearance of the diffraction at 25.4° (asterisk) characteristic of anatase (inset of Fig. 2a). XRD and XPS of other MOF samples can be found in Supplementary Figs. 5–7.

are summarized in Fig. 2a for 10%NH₂-MIL. Importantly, heating up to 300 °C has neither affected the crystalline structure of the MOFs noticeably, nor the average size and morphology of the MOF particles (Supplementary Fig. 6 and Table 3). Note that the appearance of the peak at 13.6° indicates the removal of DMF, in accordance with the simulated XRD pattern (Supplementary Fig. 2b). Further heating to 400 °C has only led to a decrease in diffraction intensity, while the characteristic pattern of the MOF structure remains largely preserved. Note that this agrees well with TGA data, which indicate the degradation of BDC-NH₂ in this temperature window (Supplementary Fig. 4). Finally, the typical MOF reflections vanished above 500 °C, which agrees with the degradation of BDC. Interestingly, the pattern at 550 °C exhibits new peaks, the most prominent being at 25.4°, which can be assigned to the anatase phase and likely correspond to tiny particles of TiO₂ formed upon complete removal of all organic species[37,38].

To examine whether or not ultrafine TiO₂ nanoclusters (~1 nm) had already been formed at lower temperatures, i.e. upon removal of BDC-NH₂, the annealed samples were further studied by TEM, SAED, Raman, and X-ray photoelectron spectroscopy (XPS). Figure 2b compares XPS Ti 2p spectra of as-prepared 10%NH₂-MIL and the samples heat-treated at 300 °C and 550 °C for 5 h. The binding energies of Ti $2p_{1/2}$ and $2p_{3/2}$ at 464.5 eV and 458.7 eV, respectively, fit very well with those reported in literature[39] confirming the presence of intact $Ti_8O_4(OH)_4$ clusters. The heat treatment at 300 °C does not affect the oxidation state, and hence the structure, in a noticeable way. This was also observed for the 5% and 50%NH₂-MIL samples (Supplementary Fig. 7). In contrast, heat treatment at 550 °C results in a considerable shift in the binding energy of 2 eV to energies characteristic of anatase-TiO₂[40]. This confirms that the transformation of 10%NH₂-MIL to TiO₂ occurs only at temperatures higher than 500 °C associated with the removal of both ligands, as indicated by XRD (Fig. 2a).

Figure 3 shows TEM images and the SAED patterns for the 5%, 10%, and 50%NH₂-MIL samples heat-treated at 300 °C for 5 h. The results reveal no noticeable changes in morphology and size of the MOF particles, in line with scanning electron

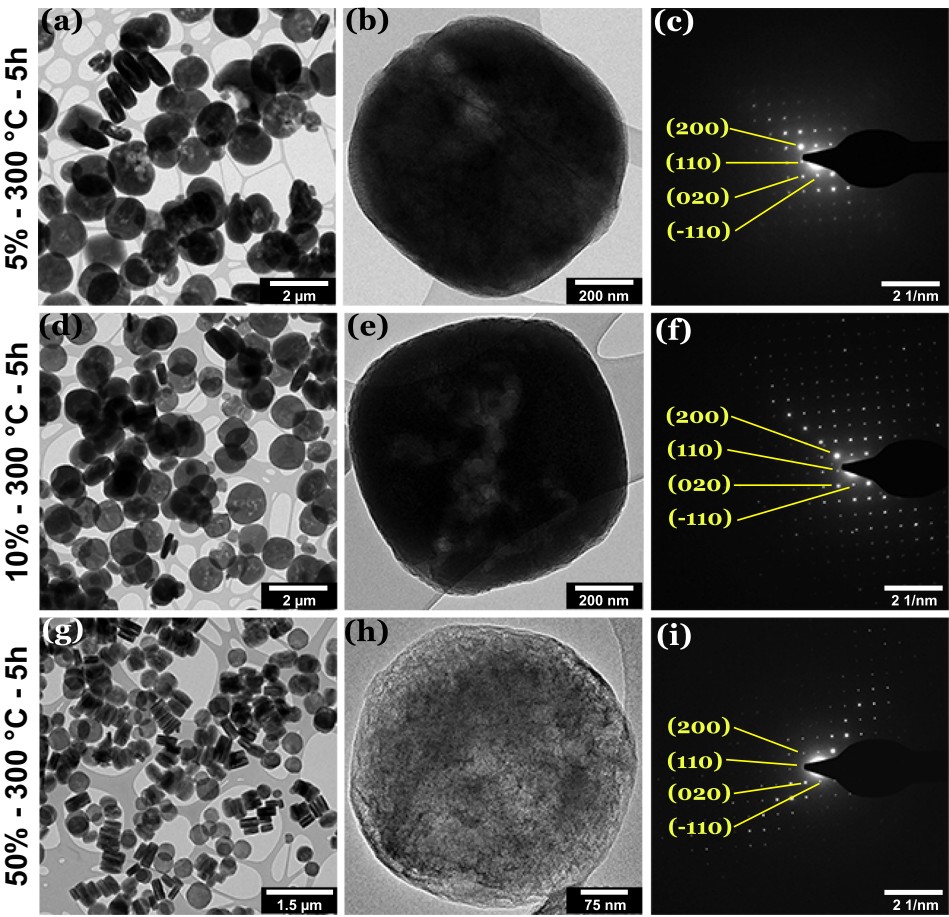

**Fig. 3 TEM/SAED analysis of MOFs heated at 300 °C for 5 h. a–c** BF-TEM images of 5%NH$_2$-MIL as an overview (**a**) and of an individual MOF particle (**b**) along with the corresponding SAED pattern (**c**). **d–f** and **g–i** show the corresponding BF TEM and SAED data for the 10% and 50%NH$_2$-MIL, respectively. All SAED patterns have been indexed using the cif file and fit the MIL-125-Ti structure,[30] viewed along the [001] zone axis. No indications of other phases were detected in the SAED data.

microscope (SEM) observations (Supplementary Fig. 6). Note that the BF images are taken from individual particles of 5% and 10%NH$_2$-MIL show some bright areas indicative of material removal. In contrast, the 50%NH$_2$-MIL samples exhibit a brighter contrast that indicates a more significant material loss that seems to affect the framework preferentially in plane perpendicular to the [001] axis (see low-magnification image in Fig. 3g). SAED confirms the preservation of the single-crystalline MIL-125 structure in all samples. Crucially, the patterns show no diffraction spots indicative of TiO$_2$ or other minority phases. Note that SAED is more sensitive towards short-range ordering than XRD[41,42], which excludes the formation of very fine clusters of TiO$_2$. In contrast, heat treatment at 550 °C (Supplementary Fig. 8) results in either transformation to anatase TiO$_2$ (e.g., 10% NH$_2$-MIL) or a complete structural damage and pore collapse (e.g., 50%NH$_2$-MIL).

These findings are supported by Raman spectroscopy (Supplementary Fig. 9). Like SAED, Raman is more sensitive to short-range ordering than XRD[42]. The spectra of as-prepared MOFs exhibit distinct peaks that can be correlated to the MIL structure[43]. Note that the MIL structure of the MOFs is largely preserved at 300 °C. Importantly, no signs of anatase-TiO$_2$[44] or other impurity phases were observed. The situation is different for samples heat-treated at 550 °C, which show features characteristic of anatase-TiO$_2$[44].

The combined results of XRD, DRIFTS, TGA/differential scanning calorimetry (DSC), Raman, XPS, and TEM/SAED thus

confirm that the heat treatment at 300 °C allows for selective removal of BDC-NH$_2$ from the mixed-ligand MOFs, while preserving the characteristic structure.

**Effect of temperature on selective ligand removal mechanism.** The selective removal of BDC-NH$_2$ from the mixed-ligand MOFs was investigated in detail with in situ DRIFTS during heating up to 600 °C. Figure 4a shows exemplary spectra of 10%NH$_2$-MIL acquired at room temperature (RT), 300 °C and 550 °C and presented for the range 2200 to 3800 cm$^{-1}$ (for the entire series of this and the other samples, refer to Supplementary Figs. 10–12). Particular interest lies in the bands between 3680 and 3600 cm$^{-1}$, which correspond to intrinsic OH groups of the Ti$_8$O$_8$(OH)$_4$ clusters[32,33]. The respective evolution plots for these bands are shown in Fig. 4b and Supplementary Fig. 11 and reveal that the heating-induced structural changes can be categorized into three thermal regimes:

In the 1$^{st}$ regime between 290 °C and 350 °C, the band at 3660 cm$^{-1}$, which is associated with OH groups of the Ti$_8$O$_8$(OH)$_4$ cluster that are H-bonded with DMF molecules (inset Fig. 4b), disappears, while a new band simultaneously arises at 3682 cm$^{-1}$. According to DFT simulations (inset Fig. 4b), this band is characteristic of the formation of free OH groups, i.e. without H-bonding. The simultaneous decrease of the CH vibrations at 2932 and 2833 cm$^{-1}$, which are associated with DMF molecules trapped within the micropores after the

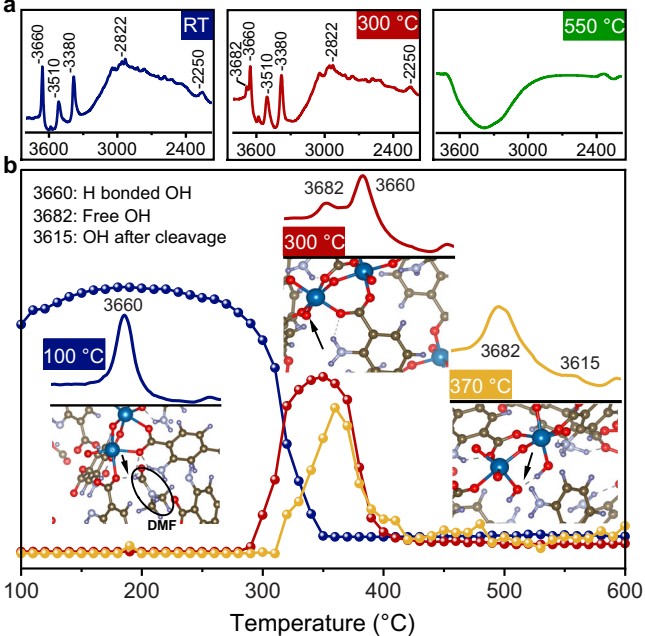

**Fig. 4 In situ DRIFTS study of 10%NH₂-MIL during heating up to 600 °C. a** Example spectra taken at room temperature (RT), 300 °C and 550 °C. **b** Selected IR bands and their corresponding thermal evolution plots. The insets show a schematic representation of the structural feature as revealed by DFT simulations: the bands correspond to OH groups that are H-bonded to trapped DMF molecules (3660 cm⁻¹), free OH groups formed after DMF removal (3682 cm⁻¹), and OH groups formed after partial or complete cleavage of the BDC-NH₂ linker from the cluster (3615 cm⁻¹).

synthesis[36], suggests that this OH-band blue-shift is a consequence of solvent removal upon thermal evaporation (Supplementary Fig. 11b). Above 350 °C, the OH-band at 3682 cm⁻¹ vanishes due to dehydroxylation of the clusters similar to other MOFs[45].

At 310 °C, a new band arises at 3615 cm⁻¹. The simultaneous decrease of the band at 1595 cm⁻¹ (Supplementary Fig. 11a), which correlates to the COO⁻ vibrations of BDC-NH₂ coordinating to the Ti₈O₈(OH)₄ cluster[6], indicate bond cleavage and the formation of new OH groups. DFT studies (Supplementary Fig. 17) confirm that the new band at 3615 cm⁻¹ is related to new OH groups at the SBU arising from bond cleavage of the SBU-BDC-NH₂ coordination (Fig. 4b). The absence of any changes of the COO⁻ vibration of the BDC coordination (1943 cm⁻¹) further confirms that bond cleavage at this temperature affects solely the BDC-NH₂ ligand.

This bond cleavage is accompanied by decarboxylation, since no bands related to C=O or C–O vibrations appeared during heating. In addition, the bands of NH and CN are largely unchanged, which suggests that at this stage the cleaved BDC-NH₂ ligands remain in the framework in the form of singly-coordinated amino species. This is supported by ¹HNMR, which confirms the presence of 2-aminobenzoic acid and excludes the formation of aniline that would be formed upon complete bond cleavage (Supplementary Fig. 15).

In the 2nd regime between 350 °C and 400 °C, the BDC-NH₂ related COO⁻ vibrations at 1595 cm⁻¹ decrease further and disappear at 410 °C, which points at the cleavage of the second ligand-cluster bond and decarboxylation to aniline. The simultaneous decrease of the CN and NH bands indicates that aniline degrades immediately after cleavage to benzene, NOₓ and water, as suggested by Feng et al[27]. This is confirmed by ¹HNMR spectroscopy (Supplementary Fig. 15). The residual weak CN/NH

peaks may be attributed to kinetic limitations associated with removing the decomposition products that are trapped in the framework, in line with XRD and TGA data. Note that the BDC characteristic CH band at 1020 cm⁻¹ remains unchanged, which documents the high selectivity of this process towards BDC-NH₂ removal.

In the 3rd regime beyond 400 °C the peak at 1020 cm⁻¹ decreases considerably at 500 °C and vanishes at 550 °C, which indicates cleavage and decomposition of the BDC ligand. This is accompanied by the disappearance of residual CN and NH peaks, which suggests the release of residual trapped NH-containing degradation products upon framework collapse. Note that the broad peak at around 3400 cm⁻¹ in the spectrum at 550 °C indicates a broad energy dispersion of OH bands likely related to the formation of TiO₂ observed in XRD and SAED.

**Isothermal ligand removal.** The MOFs were heated up to 300 °C and kept at that temperature for 20 h. We conducted in situ XRD studies of 2%, 10%, and 50%NH₂-MIL, which indicated no changes in the crystal structure upon dwelling at 300 °C (Supplementary Fig. 13).

The DRIFTS spectra of 10%NH₂-MIL for different durations upon dwelling at 300 °C, along with the temporal evolution plots of selected IR bands, are shown in Fig. 5 (for 50%NH₂-MIL refer to Supplementary Fig. 14). Within the first 60 min, the vibrations at 2932 and 2833 cm⁻¹ related to DMF disappear, and simultaneously the H-bonded OH groups (3660 cm⁻¹) transform to their free form (3682 cm⁻¹) due to solvent evaporation. Note that the main framework is still intact at this stage, as we observed no significant changes in bands intensity of the amino groups, i.e., 1252 cm⁻¹ (CN) and 3380 cm⁻¹ (NH). This is supported by ¹HNMR spectroscopy (Supplementary Fig. 15), which reveals that the typical features expected from BDC (8.04 ppm) and BDC-NH₂ (7.77, 7.39, 7.02 ppm)[46] are present in the samples before and after heat treatment for 1 h without noticeable changes in intensity.

After about 1 h, the new OH band at 3615 cm⁻¹ associated with bond cleavage of the BDC-NH₂ coordination appears. Simultaneously, the BDC-NH₂ related COO⁻ band (1955 cm⁻¹), NH-band (3380 cm⁻¹), and CN-band (1252 cm⁻¹) decrease, while the BDC-related bands at 1943 and 1020 cm⁻¹ remain largely unchanged. This demonstrates a bond cleavage of only the BDC-NH₂ linker. Note, however, that, in contrast to ligand removal upon raising the temperature, where the COO⁻ band decreases noticeably before the NH-band (Fig. 4b), the decrease in the COO⁻ and NH bands in the isothermal process occurs nearly at the same time (Fig. 5b). This suggests that the second bond cleavage starts well before the completion of the first. Again, this can be confirmed by ¹HNMR, where the spectrum taken after 5 h shows no signs of BDC-NH₂ and only small amounts of 2-amino-benzoic acid species (7.92, 7.25, 7.15, 6.91 ppm) with most of them having proceeded to degrade into gaseous NOₓ, benzene, and water[27]. Finally, cleavage, decarboxylation, and degradation of the BDC-NH₂ linker is completed after about 6 h. The situation is very similar for the 50%NH₂-MIL sample (Supplementary Fig. 15); note, however, that the cleavage of BDC-NH₂ (3615 cm⁻¹) appears to start earlier, while the complete removal takes more time compared with the 10% NH₂-MIL. Crucially, the BDC-related features in DRIFTS and NMR spectroscopy remain unchanged in all samples even after 20 h of heat treatment. This confirms the high selectivity towards removal of BDC-NH₂.

**Porosity and specific surface area.** The effect of selective linker removal on the porosity and pore architecture of the MOFs was

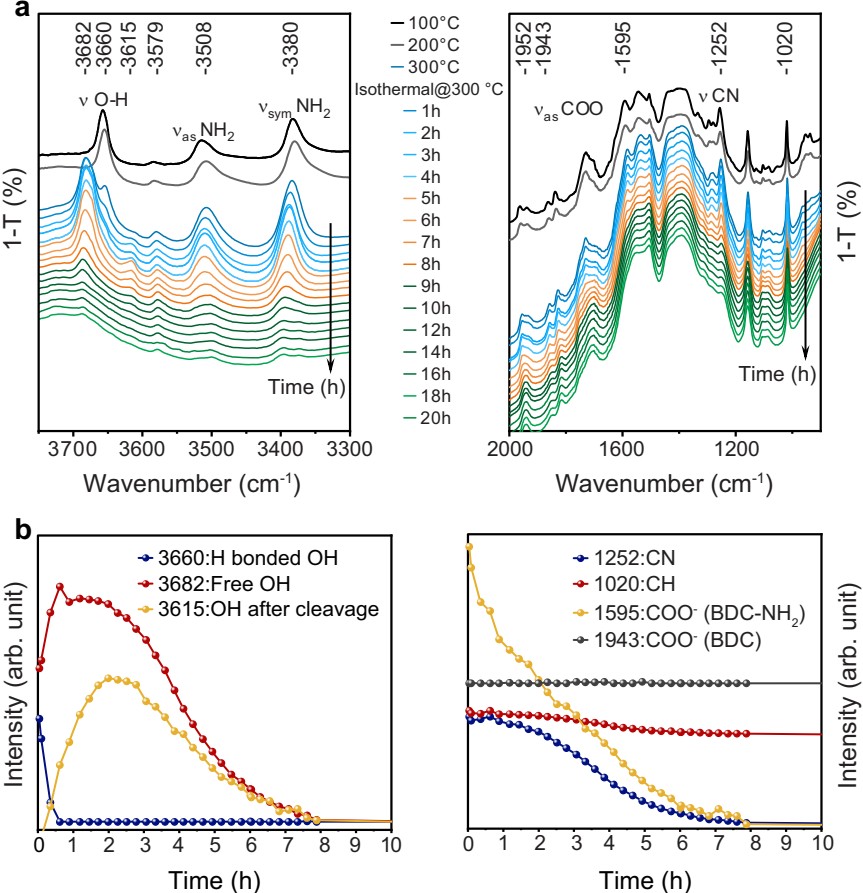

**Fig. 5 Isothermal in situ DRIFTS study of 10%NH₂-MIL. a** DRIFTS spectra were collected during heating up to 300 °C and dwelling for 20 h. **b** Evolution of characteristic IR bands during the dwelling.

investigated by physisorption with argon at 87 K (Fig. 6). The apparent surface area of the mixed-ligand MOFs decreases with increasing BDC-NH₂ content from 1400 m² g⁻¹ to ~1200 m² g⁻¹. As-prepared mixed-ligand MOFs exhibit a type I(a) isotherm typical of microporous materials with pores below 1 nm (Fig. 6a, d). The plots also show a small H4 hysteresis loop indicative of the presence of intercrystalline porosity, possibly stemming from an incomplete fusing of nuclei upon crystallization[47]. Furthermore, all samples exhibit a sharp and narrow pore size distribution (Fig. 6b, e and Supplementary Fig. 16), centered at 0.6 nm, thus confirming the high quality of the samples. Heat treatment at 300 °C leads to a diminution of the total amount of micropores and a noticeable decrease of the BET apparent surface area (Supplementary Table 5). The isotherms exhibit hybrid type I(a)/IV(a) with a hysteresis loop at higher pressures, which confirms the introduction of mesopores.

The effect of heat treatment on the porosity of MOFs, however, strongly depends on the initial BDC-NH₂ content. The samples with low BDC-NH₂ contents (2%, 5%, and 10%NH₂-MIL) experience a strong decrease in apparent surface area by about 40–50%, which is accompanied by a substantial hysteresis loop of type H5 associated with a pore structure containing both open and partially blocked mesopores. The visible tailing in the desorption curve indicates that a small portion of the network is affected by pore blocking and/or cavitation effects (Fig. 6 and Supplementary Fig. 16)[47]. As a first approximation, such pores can be seen as large cavities, with pore size distribution from 5 to 10 nm according to NLDFT. In contrast, the isotherms of the 50% and 80%NH₂-MIL samples show only a slightly reduced porosity of roughly 10–15%, which indicates that the introduced

mesoporosity is clearly less significant in terms of volume. The presence of a wide H4 hysteresis with the characteristics of cavitation-induced desorption reveals that these mesopores have access to the surface via entrances smaller than 4–5 nm. It, therefore, indicates a network of connected narrow, fracture-type mesopores. The proportion of large cavities is almost negligible for all samples with high BDC-NH₂ content (Fig. 6 and Supplementary Fig. 16) and decreases with increasing linker amount. Importantly, in all cases, the PSDs still show the presence of micropores with the same pore width of 0.6 nm as in the as-prepared samples. The combined results confirm that the selective ligand removal process at 300 °C retains the inherent microporous structure and introduces additional porosity through either cavity-type or fracture-type mesopores at low (2–10%NH₂-MIL) and high BDC-NH₂ contents (50–80%NH₂-MIL), respectively.

We performed DFT calculations of several MIL framework models and followed their structural integrity after a part of the ligands was removed simulating the carboxylate-cleavage mechanism. Figure 7 exemplarily shows the basic unit cell of MIL that has 0, 4, and 6 out of 12 ligands removed (details are given in Supplementary Fig. 18). Our models indicate that only a negligible structural distortion takes place when four or fewer ligands are removed (this corresponds to 33% of all ligands), which is in line with the structural flexibility of such frameworks[48]. In contrast, when six or more ligands are removed (corresponding to more than 50% of all ligands), the calculated distortions become significant, yet the physisorption data show that the inherent microporosity is largely preserved. As mentioned, in situ XRD shows no significant changes in

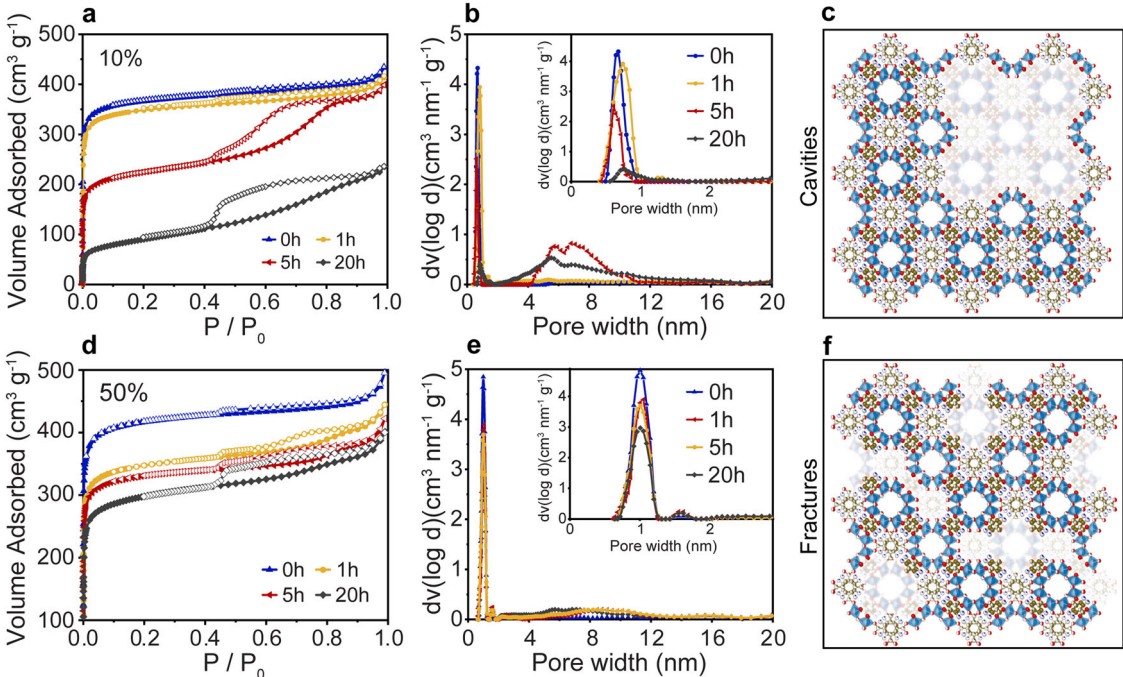

**Fig. 6 Characterization of added mesopores. a**, **d** Ar physisorption isotherms at 87 K for 10% and 50%NH$_2$-MIL respectively, for as-prepared samples and those heated at 300 °C for 1, 5 and 20 h. **b**, **e** The corresponding NLDFT pore size distributions (adsorption branch) for 10% and 50%NH$_2$-MIL. **c**, **f** Schematic representations of the mesopores reflecting cavity-type pore structures and fracture-type pores, respectively.

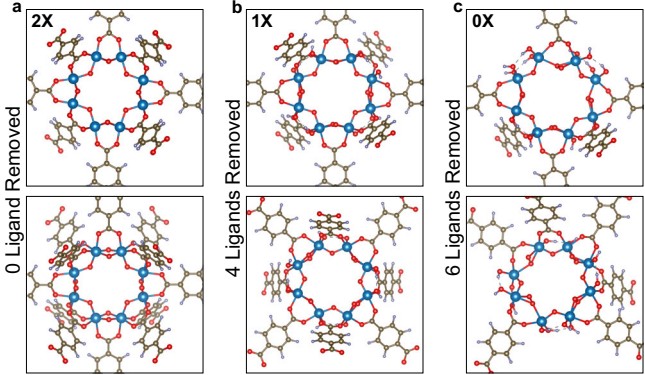

**Fig. 7 DFT simulations.** Schematic representations (top view and tilted) of the MIL frameworks with 0 (**a**), 4 (**b**) and 6 (**c**) ligands out of 12 ligands removed. A slight distortion of the pores can be seen in case c.

structure, even for 80%NH$_2$-MIL (Supplementary Figs. 12 and 13). This apparent discrepancy is likely due to the assumption that the two ligands in the frameworks are homogeneously distributed. However, the formation of mesopores upon selective ligand removal of BDC-NH$_2$ suggests a certain degree of heterogeneity and ligand clustering[49]. It is thus plausible that the resulting frameworks, even originating from 80%NH$_2$-MIL, do not experience the simulated distortion.

**Photocatalytic evaluation**. The photocatalytic properties of all as-prepared samples, as well as those heated at 300 °C for 1, 5, and 20 h, were investigated towards sacrificial hydrogen generation (HER) from aqueous methanol solutions, using a dedicated flow-reactor and a product monitoring system that allows for on-line detection of H$_2$ and CO$_2$ from the start of the process. The relatively large band gap of MIL of 3.6 eV mandates the use of a

UV light source for fair activity comparison[50]. Pt was added as a co-catalyst via in situ photodeposition[51,52]. TEM images reveal that the Pt nanoparticles are predominantly located on the surface of the MOF particles and that neither their average size of ~2 nm nor their dispersion has noticeably changed during the HER reaction (Supplementary Fig. 19). In addition, XRD and TEM also show no significant changes in the structure after 3 h of HER; note that only the 10%NH$_2$-MIL sample that was heated at 300 °C for 20 h shows some minor signs of corrosion (Supplementary Fig. 20).

The as-recorded H$_2$ evolution plots are summarized in Supplementary Fig. 21a–d and reveal that in all samples the rates reach saturation within 20–30 min and remain stable for at least several hours. Figure 8a, b compares the saturation rates - taken after 3 h of HER - for as-prepared and heat-treated 10% and 50%NH$_2$-MIL (for the other samples refer to Supplementary Fig. 22). It is clear that the heat treatment has a pronounced effect on the HER rates. The samples with 10% or lower BDC-NH$_2$ contents heated for 1 h show a slightly lower rate than the as-prepared samples. Note that heating at this stage has only removed DMF solvent molecules. In contrast, the samples 50% and 80%NH$_2$-MIL show a small increase in rate. This may be connected with an earlier start of BDC-NH$_2$ removal in those samples, as indicated by the earlier formation of OH bands (3615 cm$^{-1}$) associated with bond cleavage (Supplementary 15b).

The rates of the heated samples (5 h) are considerably higher than those of the as-prepared samples. Note that there is a major difference in the extent of rate enhancement that seems to be linked to the type of porosity. The samples with cavity-type pores (2–10%NH$_2$-MIL) exhibit a rate increase of about 1.4–1.5 times, while those containing interconnected fracture-type pores (50% and 80%NH$_2$-MIL) experience a much more pronounced 4–5 times enhancement. Changes in size and location of the Pt co-catalyst between the samples appear too small (Supplementary Fig. 19) to impact the HER rate to this extent. A contribution by TiO$_2$ nanoclusters smaller than the detection limit (<1 nm) seems

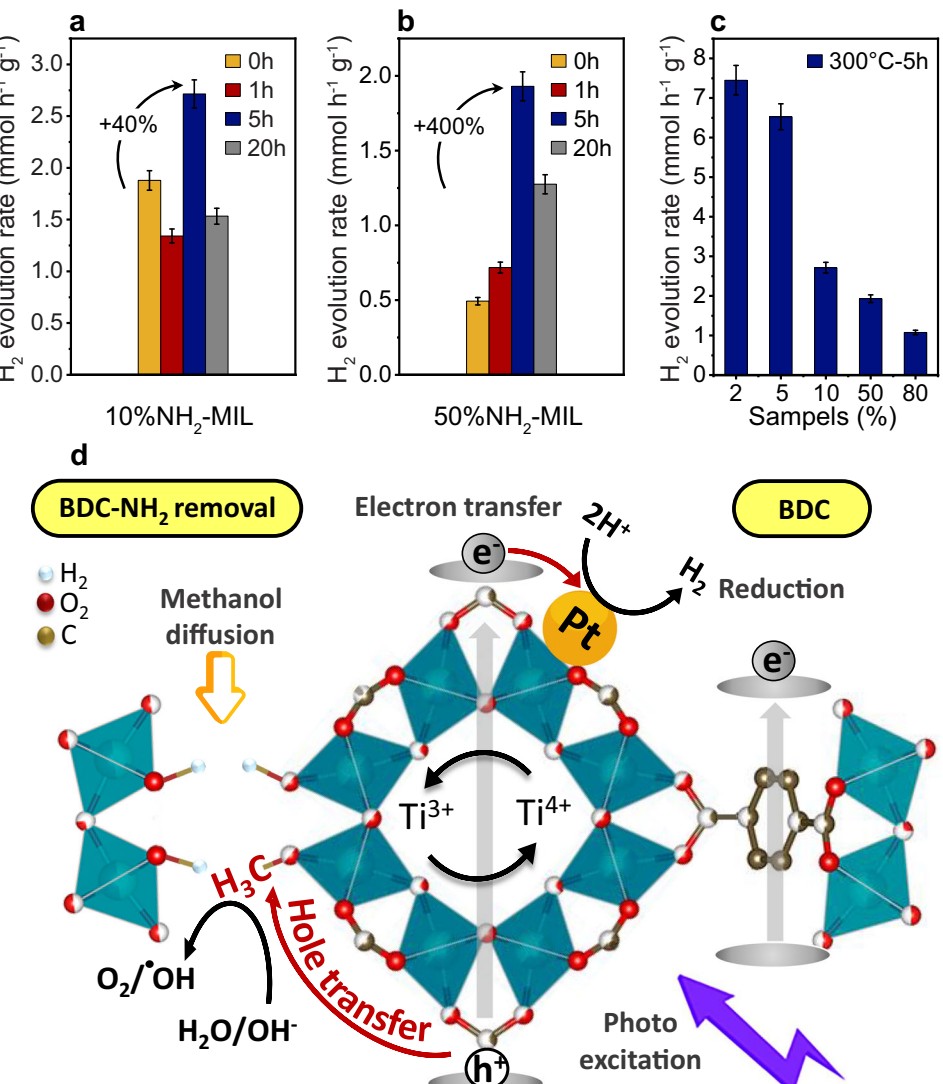

**Fig. 8 Photocatalytic studies and mechanistic models. a, b** Photocatalytic hydrogen evolution rates after 3 h HER reaction for 10% and 50%NH$_2$-MIL as-prepared and heat-treated MOFs at 300 °C for 1, 5, and 20 h. **c** Photocatalytic hydrogen evolution rates for all samples 2–80%NH$_2$-MIL heat-treated MOFs at 300 °C for 5 h. **d** Proposed mechanism for photocatalytic HER on MOFs after selective ligand removal.

unlikely in light of the complementary characterization methods, but cannot be fully excluded and requires future investigations.

The apparent correlation between rate enhancement and type of mesopores suggests that the formation of additional active sites upon selective ligand removal (i.e., terminal OH groups in the SBU) and their enhanced accessibility via reactant diffusion through large, preferably 3D-interconnected mesopores are key to the enhanced performance. This hypothesis is supported by the increased formation of CO$_2$ in the heat-treated samples (Supplementary Fig. 23), where the sacrificial agent methanol can more easily enter the pores, adsorb and get oxidized at the newly created OH groups. It further complies with the observed decrease in rate with samples treated for 20 h, which is explained by surface corrosion leading to a partial pore collapse and blockade of active sites (Supplementary Fig. 20). In addition, a more efficient charge separation at the newly formed active sites (hole trapping via methoxy formation) as well as changes in electronic structure of the MOFs, as revealed by diffuse reflectance spectroscopy (DRS) (Supplementary Fig. 25b), will likely affect the photocatalytic performance as well. Although these observations seem quite conclusive, future studies are required to investigate the effects of selective ligand removal on the band positions and defect structure, the nature of active Lewis sites[53] and the charge separation dynamics in order to fully understand their role in enhancing the photocatalytic activity in heat-treated MOFs.

In recent work, we reported that the presence of BDC-NH$_2$ is detrimental to the photocatalytic performance[50]. Essentially, photoexcitation of the organic ligand leads to electron transfer to Ti sites of the SBU, leaving the respective holes localized either at N in the ligand (for NH$_2$-MIL), or at O in the SBU (for MIL). During the reaction, methanol adsorbs on Ti as methoxy groups that are capable of scavenging nearby holes, i.e., those from a neighboring O in MIL, while the nitrogen-localized holes in NH$_2$-MIL are too far away for effective scavenging (Supplementary Fig. 26). Figure 8c shows that rates of the heat-treated MOFs decrease with increasing amino content. This is attributed to the presence of small amounts of residual BDC-NH$_2$ (or degradation products) after ligand removal, in line with DRIFTS and NMR spectroscopy. Consequently, although the 2% and 5% NH$_2$-MIL samples already outperform MIL and other photo-active MOFs[54,55], it is likely that a more complete elimination of

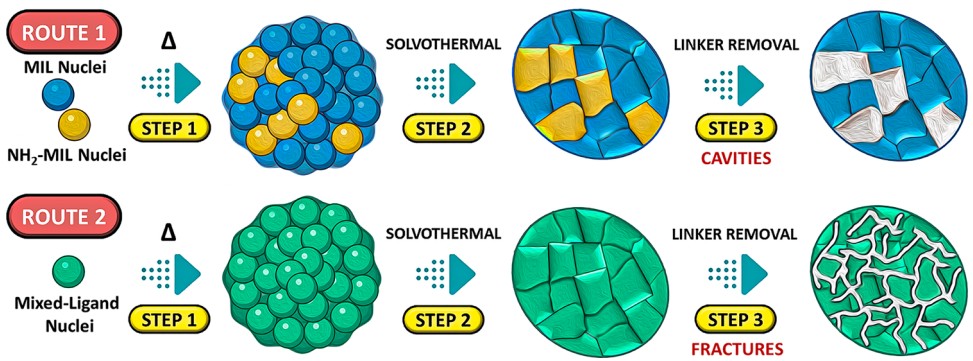

**Fig. 9 Mechanistic model of the MOFs formation.** Proposed mechanism of the MOFs formation via solvothermal for samples 2-10%NH$_2$-MIL and 50-80% NH$_2$-MIL, ranging from the formation of amorphous nuclei to crystal with selective ligand removal.

BDC-NH$_2$ via process optimizations will further enhance the HER performance.

**Hypothesis on the formation of different pore structures.** The results show that the pore structure and hence the synthesis route has a marked effect on the pore structure and the resulting enhancements in HER activity. The selective ligand removal from the 2% and 10%NH$_2$-MIL samples produced via route 1 (BDC-NH$_2$ added to the BDC-TTIP solution) yields cavity-type pores with uniform diameters that increased the activity by 1.4–1.5 times while it introduces branching fracture-type pores to the MOFs obtained via route 2 (BDC added to the BDC-NH$_2$-TTIP solution) resulting in much larger enhancements of about 5–6 times. Most works to date used post-synthetic ligand exchange to create mixed-ligand MOFs, which either resulted in the coexistence of randomly sized areas containing only single-ligands (i.e., fast ligand diffusion) or in the formation of distinct core-shell particles (i.e., slow ligand diffusion)[56]. However, none of these morphologies can satisfactorily explain the observed pore structures, nor the different enhancement factors regarding the HER activity.

In contrast to previous studies, our synthesis approach incorporates both ligands in solution at room temperature before solvothermal crystallization, yet with different orders of ligand addition (Supplementary Fig. 1). Since we observed different pore structures of the heat-treated MOFs for the two routes, the order of ligand addition must have an impact already at the stage of nucleation. To develop this hypothesis, we need to consider the binding strength of both ligands with the SBU. Therefore, we performed DFT simulations of MIL and NH$_2$-MIL frameworks bond lengths of the ligand-cluster bindings in both cases. Supplementary Figure 24 shows that the ligand coordination to the SBU in MIL is symmetric and yields an O–Ti bond length of 1.941 Å. In contrast, BDC-NH$_2$ asymmetry provides coordination with two O–Ti bond distances. The amino group can interact with one of the four oxygens of the two carboxylates, which weakens the effective O charge. As a result, this particular O–Ti bond yields a longer O-Ti distance of 1.987 Å. However, the length of the three remaining O–Ti bonds is 1.946 Å (longer than for the MIL case) due to the electronic effects of the NH$_2$ on the benzene ring. Overall, this indicates that BDC has stronger binding energy within the framework than BDC-NH$_2$.

We thus propose a mechanism (Fig. 9), where the addition of the metal precursor to the ligand solution initiates the formation of small, albeit amorphous nuclei that already resemble the ligand connectivity to the SBUs of the final MOFs. Due to the low solubility of both ligands in the DMF-methanol mixture, nucleation may be the dominant process at low temperatures.

In route 1, where BDC-NH$_2$ is added to a pre-mixed BDC-TTIP solution, most nuclei likely comprise of pure BDC-SBU, while the BDC-NH$_2$ either attaches as shells or, more likely, forms separate BDC-NH$_2$-SBU nuclei that fuse together during the solvothermal process. It is thus plausible that the MOF particles in this process are essentially assemblies of the initial, pure-ligand nuclei, which connect upon crystallization. Upon heat treatment, the parts resembling the BDC-NH$_2$ nuclei are removed preferably, resulting in cavity-type pores in the MOF particle.

In route 2, the addition of BDC to BDC-NH$_2$-SBU nuclei enacts a partial ligand exchange due to stronger interactions between BDC and the SBU, thus creating mixed-ligand nuclei already at low temperatures. The solvothermal process then leads to aggregation and subsequent crystallization into mixed-ligand MOF particles with the BDC-NH$_2$ ligand being relatively well-dispersed throughout the entire framework. Upon heat treatment, the selective removal of BDC-NH$_2$ ligands thus yields extended, but narrow fracture-type pores that are, at least partially, interconnected throughout the MOF particle.

In both cases, the overall framework structure remains preserved upon heat treatment, as confirmed by XRD, physisorption, and DFT simulations, while XPS, SAED, and Raman exclude the formation of TiO$_2$ nanoparticles. The type of generated mesopores then plays a dominant role in the photocatalytic enhancement, and it appears that the fracture-type mesopores (50% and 80%NH$_2$-MIL) with their 3D pore connectivity (i.e., improved reactant diffusion) and larger surface areas (i.e., more active sites) show stronger improvements than the cavity-type larger mesopores (2%, 5%,10%NH$_2$-MIL).

In conclusion, we produced a series of photoactive MOFs of the MIL family containing dual-porosity using a highly selective ligand removal process from mixed-ligand MOFs. Careful adjustment of synthesis conditions and heating parameters yielded either isolated cavity-type mesopores of uniform diameter (2–10%NH$_2$-MIL) or branched, narrow fracture-type pores (50–80%NH$_2$-MIL), without the typically associated collapse of the micropore structure. We analyzed the ligand removal process thoroughly with a large set of ex situ and in situ techniques, including XRD, DRIFTS, TGA/DSC, SEM, SAED/TEM, XPS, Raman, Ar physisorption, and $^1$HNMR spectroscopy, supported by DFT simulations. The ligand removal is highly selective towards BDC-NH$_2$ and follows a 2-step process that can be well controlled by adjusting temperature and time. The results show that the inherent framework remained intact, even after the removal of >50% of the ligands with no visible signs of any oxide nanoparticle formation. This is explained by a certain degree of heterogeneity in the inherent ligand distribution that is expected from the observed formation of mesopores upon heat treatment.

The introduction of both types of mesopores has greatly enhanced the photocatalytic properties of the MOFs towards hydrogen evolution. Note that it is the fracture-type geometry with its mesopore connectivity that has enhanced the activity by unprecedented levels (up to 5-times with respect to the as-prepared mixed-ligand MOF). We attribute this mainly to the formation of new active sites, their enhanced accessibility via reactant diffusion through the interconnected mesopores and an improved charge separation efficiency. It remains for future studies to evaluate the impact of selective ligand removal on changes in band positions and inter-band gap states as well as on the nature of Lewis sites and their role as adsorption sites and charge recombination centers. Therefore, this work provides an intriguing example for rationalized pore engineering in MOFs to design model systems for fundamental studies and to advance their applications in liquid reaction media including photocatalysis, photovoltaics, electrocatalysis, and electrochemical sensing.

## Methods

**Synthesis of mixed-ligand MOFs**. A series of mixed-ligand MOFs was prepared by applying a certain amount of both ligands (BDC and BDC-NH₂) to yield a series of samples with BDC-NH₂ mol.% equal to 2, 5, 10, 50, 80. In a single synthesis, addition of TTIP (0.592 mL), BDC (1000 mg for 2%, 5%, 10% and 249 and 99.8 mg for 50% and 80% respectively), BDC-NH₂ (22.2, 57.2, 120.8, 271.7, 434.7 mg, respectively), to a solution of DMF and methanol (20 mL for 2%, 5%, 10% and 40 mL for 50%, 80%, $V_{DMF}/V_{methanol}$ = 9:1) was poured in a 100 mL Teflon-lined steel autoclave. The mixture was heated at 150 °C for 18 h (2%, 5%, and 10%) and 24 h (50%, and 80%). As the nondominant linker is dispersing in the matrix of dominant linker[49], there are 2 different routes to synthesis mixed-ligand MOFs: route 1 is based on adding BDC-NH₂ to the MIL-125-Ti precursor and route 2 is adding BDC ligand to NH₂-MIL precursor. The samples with 2%, 5%, and 10% of NH₂ content were synthesized by route 1, and route 2 was applied to prepare 50% and 80%. After cooling to room temperature, the resulting powder was washed with DMF three times and twice with methanol to remove the unreacted ligand and separated by centrifugation and dried under vacuum at 150 °C for 1 day.

**Heat-treated MOFs**. Two protocols followed: (1) The samples were heated in air with a constant ramp (10 °C min⁻¹) up to 550 °C. (2) The samples were heated to 300 °C with a constant ramp (10 °C min⁻¹) and kept at that temperature for durations up to 20 h. In both cases, the samples were allowed to cool down to room temperature.

**Argon physisorption isotherms**. Physisorption isotherms were measured on a Quantatec iQ2 instrument equipped with the cryosync accessory using Ar at 87 K. The surface area was quantified via BET following the procedure recommended for microporous sorbents[57]. The pore size distributions were calculated by applying the kernel of (metastable) nonlocal density functional theory (NLDFT) adsorption isotherms.

**X-ray Diffraction**. XRD profiles were recorded on a PANalytical X'Pert Pro multi-purpose diffractometer (MPD) in Bragg Brentano geometry operating with a Cu anode at 45 kV, 40 mA, equipped with a BBHD Mirror and an X-Celerator multichannel detector. All measurements were conducted with a Cu sealed tube Kα and Kβ radiation (2:1 ratio) source (λ = 1.5406 Å) at a scan rate of 0.5° min⁻¹.

**Scanning electron microscope**. SEM images were acquired on a FEI Quanta 250 FEG SEM with an acceleration voltage of 10 KV.

**Transmission electron microscope**. TEM BF images and SAED patterns were recorded on a FEI Tecnai F20 Transmission Electron Microscope (TEM) with an electron acceleration voltage of 200 kV.

**Raman spectroscopy**. Raman measurements employed Horiba Jobin-Yvon Lab-RAM 800HR and were carried out using a 532 nm wavelength green laser.

**Crystallographic phase analysis**. Qualitative phase analysis was carried out by comparison to literature and to the International Centre for Diffraction Data (ICDD) using PDF4+ 2021 software. The phase analysis of mixed-ligand MOFs based on MIL-125-Ti MOF was carried out using the cif file provided in reference[30]. The following PDF files were used for phase identification of anatase TiO₂: 04-011-0664.

**X-ray photoelectron spectroscopy**. XPS spectra were acquired in a UHV chamber (base pressure < 3 × 10⁻¹⁰ mbar) equipped with a Specs XR50© high-intensity non-monochromatic Al/Mg dual anode and a Phoibos 100© hemi-spherical electron energy analyzer with multichannel plate detector[58].

**UV−Vis diffuse reflectance spectra**. UV-Vis spectra were measured on a Jasco V-670. The light was collected with an Ulbricht-sphere and the incident light was in the range of 300–800 nm.

**In situ XRD**. In situ XRD was measured on a PANalytical, X′Pert Pro MPD diffractometer system, conducted with Cu-Kα1,2 radiation (λ = 1.54060 Å, 1.54439 Å) and equipped with an X-Celerator multichannel detector using Bragg Brentano geometry. The samples were heated with an Anton Paar HTK 1200 N oven.

**In situ diffuse reflectance for infrared fourier transform spectroscopy**. DRIFTS data were measured on IR Tracer-100 under air atmosphere. The instrument was equipped with a controlled heating device.

**Thermogravimetric analysis**. TGA measurements were conducted on a Perkin-nElmer. Thermogravimetric analyzer 8000, using aluminum oxide (Al₂O₃) crucible.

**Differential scanning calorimetry**. DSC was performed at a heating rate of 10 °C min⁻¹ under air atmosphere with a 20 mL min⁻¹ flow rate over a temperature range of 50–600 °C.

**Nuclear magnetic resonance spectroscopy (¹HNMR)**. The liquid phase ¹H spectra were recorded on a Bruker AVANCE 250 (250.13 MHz) equipped with a 5 mm inverse-broad probe head and z-gradient unit.

**Density-functional theory**. DFT calculations were performed by using the Vienna ab initio simulation package (VASP 5.4.1) under spin-polarized density functional theory (DFT). Detailed information are given in Supplementary Information.

**Photocatalysis experiments**. We used a home-made continuous-flow reactor equipped with an online gas detection system from Emerson (X-Stream) that simultaneously monitors H₂, O₂, CO, and CO₂. As a light source a 200-W super-pressure Hg lamp (λ = 240–500 nm, I = 30mWcm-2, Superlite SUV DC-P deep UV, Lumatec) equipped with an optical fiber light guide was used.

## Data availability

Additional data on methods, materials characterizations and photocatalytic performance are available in Supplementary Information. Source data are provided on this platform (10.5281/zenodo.5727081).

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

## Acknowledgements

This research was funded in whole, or in part, by the Austrian Science Fund (FWF, I 5413-N, S.N.). The authors acknowledge the use of facilities at the University Service Centre for Transmission Electron Microscopy (USTEM) and the X-Ray Center (XRC), both located at the TU Wien, as well as financial support of the Vienna University of Technology, the University of Vienna, as well as via the Nancy and Stephen Grand Technion Energy Program (GTEP, upon work from COST Action CA18234, supported by COST (European Cooperation in Science and Technology), M.C.T.). Shaghayegh Naghdi acknowledges the valuable assistance and discussion from Mohammad Zendehbad from the Department of Water, Atmosphere and Environment, University of Natural Resources and Life Sciences Vienna (BOKU), Austria.

## Author contributions

S.N.: literature survey, methodology, main data acquisition and analysis, and manuscript draft writing. A.C.: discussion of photocatalysis and manuscript draft writing. A.G.: TEM experiments and DRIFTS data analysis. J.W.: additional SEM experiments. R.G.N. and F.K.: gas adsorption–desorption studies. T.G. and B.C.B.: TEM/SAED analysis. T.G.: Raman spectroscopy. T.H. and G.R.: XPS measurements and analysis. S.B. and M.C.T.: DFT calculations. D.E.: conceptualization, supervision, project administration, funding acquisition, and manuscript writing. All authors have approved the manuscript.

## Competing interests

The authors declare no competing interests.
