## [Peer Review File · Nature Communications]

Selective ligand removal to improve accessibility of active sites in hierarchical MOFs for heterogeneous photocatalysisREVIEWER COMMENTS

Reviewer #1 (Remarks to the Author):

The manuscript by Eder et al. reports the effect of thermal treatments on mixed ligand MIL-125 Ti on the structure of the resulting materials and on their photocatalytic performance. The authors claim that it is possible to selectively remove NH₂ functionalized linkers while keeping the structure of the MOF intact and that the improved transport properties of the resulting material are responsible for the observed enhanced photocatalytic performance.

This is an interesting concept, however, from reading the manuscript I could not find any experimental result that clearly demonstrates the hypothesis of ligand removal without damaging the MOF structure. MOFs have been shown in the literature to decompose upon thermal treatment, resulting in the formation of small nanoparticles (TiO₂ in case of MIL-125). I don't think that the techniques used by the authors to characterize the material can exclude the formation of TiO₂, which, in my opinion, is responsible for the enhanced photocatalytic performance, as already demonstrated before.

Following this, I cannot recommend publication unless the authors demonstrate the main claim by i.e. PDF, Ti NMR or EXAFS.

In addition to this major point, I also wonder about the selection of figures to be present in the supplementary information and main text. To the opinion of this referee, Figures 1-4 could very well be placed in the SI (this is all well known information and already reported in a large number of publications), while other info such as, UV-VIS, TEM (of better quality than the one shown in the SI) and others, would be much better suited for the main text.

Last but not least, the authors should avoid presenting XRD patterns using the selected 3D representation (in Figures 2, 3 and others), although fancy, this type of representations are not adequate for a scientific publication, where the reader should be able to extract as much information as possible from the represented data.

Reviewer #2 (Remarks to the Author):

This is an interesting paper on the process of removing ligands in MOFs to make active sites more accessible. The characterization of the missing-ligand defects is – in my view – the most interesting aspect of the paper. This makes the paper interesting even though the MOF has too large of a band gap to be interesting for photocatalysis.

Two suggested improvements:

1. The authors focus entirely on the improved diffusion in the defected MOF. I think they should mention that missing ligands can also change the nature of the catalytic sites and hence the power. A suitable reference would be: "Copper-Zirconia Interfaces in MOF UiO-66 Enable Selective Catalytic Hydrogenation of CO₂ to Methanol," Y. Zhu, J. Zheng, J. Ye, Y. Cui, K. Koh, L. Kovarik, D. M. Camaioni, J. Fulton, D. G. Truhlar, M. Neurock, C. J. Cramer, O. Y. Gutiérrez, and J. A. Lercher, *Nature Communications* 11, 5849/1–11 (2020). doi.org/ 10.1038/s41467-020-19438-w
2. The authors provide INCAR files and pictures of the structures, which is very good. However, they should also say which density functional is used, and they should give coordinate files for optimized structures.
3. The authors should explain or define the meaning of the term “multi-modal”, which appears two places (with different spellings).
4. The authors say in the conclusions that “this yields has greatly enhanced the photocatalytic properties of the MOFs towards hydrogen evolution, yielding some of the currently most-active photocatalytic MOFs.” There is no literature review that shows that this is among the “currently most-active” photocatalysts. furthermore, I do not know what they mean by “most active”. This claim needs to be justified or removed.

Reviewer #3 (Remarks to the Author):

In this work, Naghdi et al. introduce a novel strategy for controlling porosity in mixed ligand MOFs via thermolysis. A detailed analysis is performed regarding structure-thermal treatment relationships; furthermore, the obtained samples are analyzed in terms of HER rate efficiency; improvements in

efficiency are linked with the obtained pore geometries. I have some concerns regarding the validity of the conclusions derived by the analysis, details are given below:

The absence of a XRD peak at 13.6 degrees is assigned to the presence of solvent species within the framework. Following figure 2a, looks like the evolution of this peak vs NH₂-MIL content in the samples presents an abrupt “jump” for concentrations above 10%. This onset seems to indicate that a different amount of solvent molecules are trapped in the samples for synthesis route I and II. The authors should discuss this effect, and their implications in the results presented within the manuscript.

Are the mixed ligands evenly distributed in the samples for both routes? Would be nice to check that, maybe mass spec could provide some relevant info, e.g. whether surface segregation of one type of ligand is preferentially present in any of the routes analyzed.

In supplementary figure 2, the peak associated with the 101 plane in XRD seems to drop in intensity substantially after 20h of 300C heat treatment. This is not clear in the way data is presented in the main text, figure 3c. A clarification is needed.

As seen in supplementary info, the onset-temperature of weight loss depends on sample recipe. It is fair to compare recipe I and II under the same 300C recipes?

Different synthesis routes provide samples with different surface area, which is reduced to a different extent upon thermolysis. The analysis suggest that different porosity is triggered in the samples. This is in part suggested from the different evolution in pore size shown in fig 5 vs time. Following my previous comment, if the onset-temperature of weight loss depends on sample recipe, it is fair to compare pore distributions for a given time upon thermolysis for both recipes? One could argue that 20h recipe for 50% sample under 300C could be equivalent to 1h recipe for the 10% sample under 300C. In other words, could be possible that both samples reveal identical evolution of pore size and architecture vs T if that T is made relative vs weight loss onset T?

Comparing photocatalytic activity for such dissimilar samples is very tricky. Authors state that they employ 3.6eV photons for a fair comparison. But are optical densities at this energy the same for the compared samples under H₂ evolution? This will provide same number of photons absorbed. But note that sample-to-sample absorption differs in lineshape (supplementary figure 25), band tailing extending to the IR for samples treated by 300C suggest defect absorption which obviously can have a large impact in H₂ evolution (plus: I miss in supplementary fig. 25 the absorption for samples with % above 10). Even if these aspects are taking care of, one could question whether the workfunctions of the materials are changing with % of NH₂-MIL. Differences in workfunction will dramatically affects photocatalytic activity. Finally, authors use a Pt co-catalyst. In SI they show changes in Pt size (diameter, radius?) from 1.9 to

2.21nm. Co-catalyst size can affect as well its own workfunction which co-determine electron transfer rates and HER efficiencies in the samples. The authors should comment on these aspects and clarify their eventual impact on the direct correlation established between structure and photocatalytic activity.

We thank all referees for their encouraging assessments and comments. In the following, we provide a point-by-point answer and a list of revisions.

Reviewer #1 (Remarks to the Author):

The manuscript by Eder et al. reports the effect of thermal treatments on mixed ligand MIL-125 Ti on the structure of the resulting materials and on their photocatalytic performance. The authors claim that it is possible to selectively remove NH₂ functionalized linkers while keeping the structure of the MOF intact and that the improved transport properties of the resulting material are responsible for the observed enhanced photocatalytic performance.

This is an interesting concept, however, from reading the manuscript I could not find any experimental result that clearly demonstrates the hypothesis of ligand removal without damaging the MOF structure. MOFs have been shown in the literature to decompose upon thermal treatment, resulting in the formation of small nanoparticles (TiO₂ in case of MIL-125).

We thank the referee for assessing our work as an "interesting concept" and understand these concerns. The previous literature has studied thermal treatment conditions, upon which MOFs indeed decompose to yield fine metal oxide clusters that often reside within carbonaceous material. This complete decomposition, however, typically occurs at considerably higher temperatures than those applied in our work. In fact, we have also demonstrated that our MIL-125-Ti samples decompose completely at about 550 °C and yield of fine TiO₂ anatase nanoparticles. This has been documented by various microscopic, spectroscopic and diffraction techniques.

However, the main purpose of our work was to develop a process that removes only one ligand with high selectivity, while preserving the overall structure of the MOFs and adding mesoporosity. This is achieved by deliberately staying below the conditions of complete thermal conversion, i.e. choosing conditions, at which only one of the two ligands gets selectively thermolyzed – our results show that this happens in the range between 300-350 °C for the MOFs studied in this work.

I don't think that the techniques used by the authors to characterize the material can exclude the formation of TiO₂, which, in my opinion, is responsible for the enhanced photocatalytic performance, as already demonstrated before.

As mentioned before, we dedicated this research to 1) confirm the preservation of the intrinsic MOF structure and microporosity, 2) evaluate the nature of the additional mesoporosity, and 3) to exclude the formation of TiO₂ nanoparticles upon selective ligand removal. Therefore, we used a very large set of cutting-edge methods to investigate these points in great detail.

- a) TGA shows distinct weight losses at roughly 300 °C and 500 °C, respectively, which correspond very well to the respective amounts of BDC-NH₂ and BDC in the as-prepared MOFs, respectively. We also performed an isothermal TGA (where the temperature was kept at 300 °C for 20 hours) that also shows the quantitative removal of BDC-NH₂. Hence, these results document that the BDC-NH₂ ligand can be removed completely and selectively upon heating in air at 300 °C.

REVISIONS: Additional TGA in Supplementary Fig. 4d.

- b) XRD shows that the heat treatment at 300 °C has not changed the crystal structure and has not led to the formation of any detectable TiO₂ species. In contrast, heat treatment at 550 °C has destroyed the structure and in fact has resulted in a weak, broad peak at 25.4° that documents the appearance of anatase nanoparticles; hence, the formation of TiO₂ nanoparticles is clearly correlated with the complete degradation of the MOFs (i.e. decomposition of both ligands).

REVISIONS: We show the most important XRD patterns in direct comparison (Fig. 2a) to better demonstrate the aforementioned observations and to highlight the formation of anatase at 550 °C. For completeness, we added a comparison of the corresponding XRD patterns at 300 °C for all samples in Supplementary Fig. 5.

- c) TEM/SAED: The TEM images and corresponding SAED patterns clearly show that the individual particles of as-prepared MOFs (Fig. 1) as well as of MOFs heated at 300 °C for 5

hours (Fig. 3) are single-crystalline in nature and correspond to the expected structure of MIL-125-Ti. The diffractions show no signs of structural changes, nor the presence of any impurity phases, such as TiO₂ nanoparticles. Since SAED is more sensitive than XRD and allows the detection of ultrafine particles (~ 1 nm), we can exclude that TiO₂ nanoparticles of 1 nm or larger were formed at 300 °C.

In contrast, the TEM images of the samples heat-treated at 550 °C (Supplementary) show a considerable degree of structural degradation. In addition, SAED now shows additional diffractions that can be assigned to TiO₂.

REVISIONS: These are NEW results that support our other data very well. TEM and SAED both confirm the preservation of the MIL structure and the absence of TiO₂ species at 300 °C, which only appear at 550 °C. The results are added in Fig. 1, Fig. 3 and Supplementary Fig. 8.

- d) Raman spectroscopy: We collected Raman spectra of the as-prepared samples and those heat-treated at 300 °C. Both show the expected features of the MIL structure as well as the absence of the Eg band at 144 cm⁻¹ (anatase) or 190 cm⁻¹ (rutile). Since Raman spectroscopy is also a highly sensitive technique, this confirms the absence of any ultrafine TiO₂ species at 300 °C.

In contrast, the Eg band for anatase appears only in the samples heat-treated at 550 °C.

REVISIONS: These are also NEW results dedicated to the questions of crystal structure preservation and TiO₂ formation. The spectra are added in Supplementary Fig. 9.

- e) XPS: The Ti 2p spectra of the as-prepared samples show binding energies at positions that agree very well with literature data for MIL-125-Ti. There was no change in position for the samples heat-treated at 300 °C for 5 hours. Only the samples at 550 °C show a clear shift by about 2 eV to positions expected for TiO₂ particles. Such a large shift allows for a clear distinction between MIL-125-Ti and TiO₂. Therefore, these results further confirm the absence of TiO₂ at 300 °C and their formation at 550 °C.

REVISIONS: These are also NEW results. The results for the as-prepared 5%, 10% and 50%NH₂-MIL samples and those heat-treated at 300 °C and 550 °C are added in Fig. 2b and Supplementary Fig. 7.

- f) DRIFTS shows that BDC-NH₂ can be selectively removed at 300 °C (completely gone after about 7 hours). In fact, we were able to identify the individual stages of a) the first bond cleavage along with decarboxylation to yield singly-attached 2-benzoic acid species, and b) the second cleavage of and decomposition into benzene, CO₂ and NO_x species. This is accompanied by the formation of new OH groups in the SBU (as supported by DFT simulations). All these OH bands are very pronounced and sharp.

In contrast, at 500-550 °C a broad OH band appears that is associated with TiO₂ formation. This is accompanied by the decomposition of the BDC ligand. Again, this documents that TiO₂ formation only occurs at temperatures above 500 °C upon complete degradation.

REVISIONS: We highlight the evolution of the characteristic OH bands in Fig. 4. Additional spectra and evolution plots were moved to Supplementary Fig. 10-11 for better clarity. The description of the DRIFTS results was rewritten more concisely.

- g) Supporting characterizations: ¹HNMR spectroscopy confirms the selective removal of BDC-NH₂ at 300 °C and the preservation of BDC in the framework. DFT simulations confirm that the removal of up to 50% of the ligands preserves the framework structure. Argon physisorption also confirms that the MOFs heat-treated at 300 °C still contain micropores of the same diameter and shape as the as-prepared MOFs. The samples heat-treated at 550 °C show a considerably reduced porosity with blocked and collapsed pores of different shapes and sizes, confirming the full degradation of the samples.

Following this, I cannot recommend publication unless the authors demonstrate the main claim by i.e. PDF, Ti NMR or EXAFS.

We agree with the referee about the need of confirmation. As detailed above, we chose, however, other techniques that we believe are better suited for answering this particular question with our

samples. The combined microscopic, spectroscopic and diffraction results clearly support our previous claims so that we can now unambiguously exclude the formation of TiO₂ nanoparticles.

The suggested methods are certainly of interest, but out of scope for this work. EXAFS and PDF studies would require synchrotron radiation to be able to identify species smaller than 1 nm, for which we would need to apply for beam time first. Moreover, the rather small amounts of Ti in the samples would impose serious limitations and challenges, in particular with solid-state Ti NMR spectroscopy.

Therefore, we are indebted to the referee, who made us think more deeply and caused us to increase our efforts to proof our claims. The new studies will provide added valuable to the entire MOF community.

Revisions:

1. We performed new experiments and added the TEM/SAED results in Fig. 1 (as-prepared 5% and 50%NH₂-MIL) and Fig. 3 (5%, 10% and 50%NH₂-MIL after heat treatment at 300 °C for 5 hours). The corresponding results on the samples heated at 550 °C are summarized in Supplementary Fig. 8. The discussion of the results is embedded in the main manuscript on pages 3 and 7.
2. We performed new experiments and added the XPS results in Fig. 2b (10%NH₂-MIL as-prepared and heat-treated at 300 °C and 550 °C for 5 hours). The corresponding data on 5% and 50%NH₂-MIL are shown in Supplementary Fig. 7. The discussion of the data can be found on page 6.
3. We performed new experiments and added the Raman data in Supplementary Fig. 9 and their discussion on page 7.
4. We revised the corresponding discussion sections for XRD and DRIFTS to emphasize the absence of TiO₂ in conjunction with the new results.

In addition to this major point, I also wonder about the selection of figures to be present in the supplementary information and main text. To the opinion of this referee, Figures 1-4 could very well be placed in the SI (this is all well-known information and already reported in a large number of publications), while other info such as, UV-VIS, TEM (of better quality than the one shown in the SI) and others, would be much better suited for the main text.

Thank you for this suggestion. We believe that most of the results previously shown in Figures 1-4 are not well-known, as very few studies have investigated the (complete) thermal conversion of mixed-ligand MIL-125-Ti. However, we agree with the referee that the results on the as-prepared mixed-ligand MOFs are not the type of highlight results that warrants prominent placement in the main manuscript.

Revisions: We restructured this manuscript substantially and moved

1. the schematics of the synthesis routes to Fig. S1
2. XRD of all as-prepared MOFs to Fig. S2 and additional heat-treated MOFs to Fig. S5
3. DRIFTS of as-prepared MOFs to Fig. S3 and Table S2
4. TGA/DSC to Fig. S4
5. SEM of as-prepared and heat-treated MOFs (300 °C) to Fig. S6 and Table S3
6. Evolution plots from *in situ* DRIFTS of 10%NH₂-MIL to Fig. S11
7. XRD of MOFs kept at 300 °C for 20 hours to Fig. S13

In addition, we added our new data obtained from TEM/SAED and XPS to the text. We believe that these revisions have greatly improved the quality and impact of this manuscript.

Last but not least, the authors should avoid presenting XRD patterns using the selected 3D representation (in Figures 2, 3 and others), although fancy, this type of representations are not adequate for a scientific publication, where the reader should be able to extract as much information as possible from the represented data.

We agree with the referee. We revised the style of representation accordingly.

Reviewer #2 (Remarks to the Author):

This is an interesting paper on the process of removing ligands in MOFs to make active sites more accessible. The characterization of the missing-ligand defects is – in my view – the most interesting aspect of the paper. This makes the paper interesting even though the MOF has too large of a band gap to be interesting for photocatalysis.

We thank the referee very much for this highly positive and encouraging assessment of our work. In particular, we are happy that our in-depth studies on the defect formation/ligand removal mechanism has been noted as “the most interesting aspect”, which we also believe.

Currently, we are employing this strategy of selective ligand removal to MOFs that are better suited for visible-light applications – with very promising preliminary results indeed. Therefore, we hope that this work will stimulate fellow researchers to use this approach to many MOFs and study their performance in many applications.

Two suggested improvements:

1. The authors focus entirely on the improved diffusion in the defected MOF. I think they should mention that missing ligands can also change the nature of the catalytic sites and hence the power. A suitable reference would be: "Copper-Zirconia Interfaces in MOF UiO-66 Enable Selective Catalytic Hydrogenation of CO₂ to Methanol," Y. Zhu, J. Zheng, J. Ye, Y. Cui, K. Koh, L. Kovarik, D. M. Camaioni, J. Fulton, D. G. Truhlar, M. Neurock, C. J. Cramer, O. Y. Gutiérrez, and J. A. Lercher, Nature Communications 11, 5849/1–11 (2020). doi.org/ 10.1038/s41467-020-19438-w

This is a very good comment and an excellent paper. We agree that there is indeed a strong need for future studies to uncover the effects of selective ligand removal on the nature of active sites and their contribution to the photocatalytic performance.

Revisions: We added this interesting paper as REF. 53 and added some comments in the photocatalysis and conclusion sections.

2. The authors provide INCAR files and pictures of the structures, which is very good. However, they should also say which density functional is used, and they should give coordinate files for optimized structures.

We used Perdew-Burke-Ernzerhof (PBE) exchange-correlation (XC) functional for our calculations and all the detailed information are given in the SI (Section 1: Density-functional theory (DFT) part). All the optimized structure coordinates are given in the additional file (.xlsx). We thank the referee for suggesting this.

3. The authors should explain or define the meaning of the term “multi-modal”, which appears two places (with different spellings).

Thank you for this comment. We changed this term to read “dual-porosity” as it better fits the description in the text.

4. The authors say in the conclusions that “this yields has greatly enhanced the photocatalytic properties of the MOFs towards hydrogen evolution, yielding some of the currently most-active photocatalytic MOFs.” There is no literature review that shows that this is among the “currently most-active” photocatalysts. furthermore, I do not know what they mean by “most active”. This claim needs to be justified or removed.

We compared the activity towards HER of our MOFs with similar MOFs in similar conditions (Pt as co-catalyst, comparable light source, similar sacrificial agents) and found that these MOFs were comparable to some of the most active catalysts. However, we understand that such direct comparisons are challenging and questionable.

Reviewer #3 (Remarks to the Author):

In this work, Naghdi et al. introduce a novel strategy for controlling porosity in mixed ligand MOFs via thermolysis. A detailed analysis is performed regarding structure-thermal treatment relationships; furthermore, the obtained samples are analyzed in terms of HER rate efficiency; improvements in efficiency are linked with the obtained pore geometries.

We thank the referee for this highly positive assessment of our work.

I have some concerns regarding the validity of the conclusions derived by the analysis, details are given below:

The absence of a XRD peak at 13.6 degrees is assigned to the presence of solvent species within the framework. Following figure 2a, looks like the evolution of this peak vs NH₂-MIL content in the samples presents an abrupt "jump" for concentrations above 10%. This onset seems to indicate that a different amount of solvent molecules are trapped in the samples for synthesis route I and II. The authors should discuss this effect, and their implications in the results presented within the manuscript.

We thank the referee for this observation, whose origin is, however, rather difficult to answer. The peak at 13.6° generally seems to decrease in intensity with an increasing amount of BDC-NH₂ in the as-prepared samples. We attributed this mainly to the increased polarity upon addition of BDC-NH₂.

Yet, we agree that the change in intensity appears rather abrupt between the 10% and 50% sample, which suggests that the order of ligand addition before crystallization has an effect on the DMF content as well. It may be that the presence of BDC-NH₂ in all mixed-ligand nuclei of route 2 attracts more DMF molecules, compared with the single-ligand nuclei of route 1, where DMF is confined to the pure BDC-NH₂ nuclei (max 10%).

Revisions: We moved the discussion of the 13.6° peak to Supplementary Information (Fig. 2b and page 9) and added a sentence to highlight this observation.

Are the mixed ligands evenly distributed in the samples for both routes? Would be nice to check that, maybe mass spec could provide some relevant info, e.g. whether surface segregation of one type of ligand is preferentially present in any of the routes analyzed.

We agree with the referee. In our view, the ligands need to be non-homogeneously distributed in order to produce mesopores upon selective ligand removal. As we also discussed in the hypothesis section, the synthesis routes enforce different ligand distributions already at the level of nuclei formation, hence resulting in different mesopore structures.

Yet, we would be very interested in a quantitative evaluation of this ligand distribution in a future collaborative study, in order to optimize our process.

In supplementary figure 2, the peak associated with the 101 plane in XRD seems to drop in intensity substantially after 20h of 300C heat treatment. This is not clear in the way data is presented in the main text, figure 3c. A clarification is needed.

In the former submission, Supplementary Fig. 2 contained *ex situ* XRD patterns taken from the samples after the respective heat treatment. We did not use an internal standard, hence, I would be careful to compare the intensities in these patterns, as they are likely affected by sample preparation.

However, we collected *in situ* XRD patterns for 2%, 10% and 50%NH₂-MIL during dwelling at 300 °C for 20 hours – these patterns were shown in Fig. 3c in the previous manuscript and are now placed in Supplementary Fig. 13. These patterns do not show significant changes in intensity, thus we argue that the overall structure of the MOFs has not been altered by the heat treatment.

Revision: We clarified this in the text and only discussed our *in situ* XRD results, which are shown in Supplementary 5 and 13.

As seen in supplementary info, the onset-temperature of weight loss depends on sample recipe. It is fair to compare recipe I and II under the same 300C recipes? Different synthesis routes provide samples with different surface area, which is reduced to a different extent upon thermolysis. The analysis suggest that different porosity is triggered in the samples. This is in part suggested from the different evolution in pore size shown in fig 5 vs time. Following my previous comment, if the onset-temperature of weight loss depends on sample recipe, it is fair to compare pore distributions for a given time upon thermolysis for both recipes?

We agree that the small differences in onset temperature observed in TGA would affect the extent of ligand removal for this process, i.e. the amount of residual amino-species, consequently affecting the photocatalytic performance, as we in fact discussed in the photocatalysis section. However, it would be very tedious to fine-tune the temperature for each sample to achieve the exact same amount of residual species.

It is also important to mention that we did not compare the different samples with each other (nor the two sample sets), but evaluated the enhancements in photocatalytic performance of the heat-treated samples compared with the corresponding as-prepared samples.

One could argue that 20h recipe for 50% sample under 300C could be equivalent to 1h recipe for the 10% sample under 300C. In other words, could be possible that both samples reveal identical evolution of pore size and architecture vs T if that T is made relative vs weight loss onset T?

This is an interesting idea. However, our physisorption data of the two sample in question, i.e. 50%/20h and 10%/1h (Fig 6), clearly show different isotherms with distinct types of hysteresis, hence proving different porosity, pore sizes and pore structures.

This shows that, while we cannot exclude small contributions by the thermolysis protocol, the mesopore characteristics in the heat-treated samples are clearly defined predominantly by the initial ligand distribution, i.e. the synthesis route and the initial BDC-NH₂ content.

Comparing photocatalytic activity for such dissimilar samples is very tricky. Authors state that they employ 3.6eV photons for a fair comparison. But are optical densities at this energy the same for the compared samples under H₂ evolution?

Note that we only compare here the performances towards HER (not the quantum efficiencies) of the samples before and after the selective ligand removal. In doing so, we have to use the same light conditions and reactor setting. It is neither practicable nor sensible to measure the optical densities and fine-tune the light conditions for each individual sample.

This will provide same number of photons absorbed. But note that sample-to-sample absorption differs in lineshape (supplementary figure 25), band tailing extending to the IR for samples treated by 300C suggest defect absorption which obviously can have a large impact in H₂ evolution. Even if these aspects are taking care of, one could question whether the workfunctions of the materials are changing with % of NH₂-MIL. Differences in workfunction will dramatically affects photocatalytic activity.

We agree with the referee. Selective ligand removal has not only led to the formation of mesopores that can facilitate reactant diffusion, but has also introduced new active sites (see "cleaved OH bands" in DRIFTS) and affected the electronic structure of the MOFs (DRS, Supplementary Fig. 25).

Therefore, we emphasize that we need future studies to evaluate the formation of Lewis sites, investigate changes in work function, charge formation, transport and recombination dynamics and assess their impact on the photocatalytic performance of MOFs.

We believe that our work will stimulate many future studies aimed at gaining more insights into the electronic properties and adsorption sites of MIL and other MOFs to advance their application in (photo)catalysis.

Revisions: We added some comments in the photocatalysis and conclusion sections

Finally, authors use a Pt co-catalyst. In SI they show changes in Pt size (diameter, radius?) from 1.9 to 2.21nm. Co-catalyst size can affect as well its own workfunction which co-determine electron transfer rates and HER efficiencies in the samples. The authors should comment on these aspects and clarify their eventual impact on the direct correlation established between structure and photocatalytic activity.

Indeed, the size of the co-catalyst and their location generally have a significant impact on the photocatalytic performance. However, in this work we compared the respective samples before and after selective ligand removal. Our TEM results confirm that the Pt sizes are the same for the respective samples before and after heat-treatment and there is also no change in size during HER reaction.

Revisions: We added a section to clarify this in the photocatalysis section (page 13).

REVIEWERS' COMMENTS

Reviewer #1 (Remarks to the Author):

I appreciate the efforts made by the authors in addressing my previous comments and paying more attention to the characterization of the different samples. Yet, the presence of TiO₂ nanoparticles smaller than 1 nm cannot be ruled out and the nature of the active sites is not clear. I am happy to see that the authors have open the door to an explanation that goes beyond diffusion to rationalize the improved photocatalytic performance, however, in my opinion, the paper is not at the scientific level that I would like to see in Nature Communications. I therefore leave it up to the Editor to decide

Reviewer #3 (Remarks to the Author):

Reading the revised paper and the answers to the reviewers, I do believe that the authors provide sufficient experimental evidence supporting their claims regarding a novel avenue to control porosity in MOFs. However, assigning the improved HER activity to a better reactant access to active sites, while plausible, seems a premature claim that I consider is not supported by the presented data. While authors has included some text in the new version stating that other factors might affect HER (gaps, defects, work-functions, Pt size), some old text remain in the current version making precisely an strong link between porosity and HER, e.g.:

Page 14 – “In absence of any changes in size and location of the Pt co-catalyst (Supplementary Fig. 19) and of any TiO₂ nanoparticles, the different enhancement factors must be linked to the different type of introduced mesoporosity.”

Page 18 – “ Importantly, it is the fracture-type geometry with its mesopore connectivity and thus better reactant access to actives sites that has enhanced the activity by unprecedented levels (up to 6-times with respect to the as-prepared mixed-ligand MOF).”

Once these claims are tone down the paper could be published in Nat. Comm.

Referee 1:

We thank the referee for reading our revised manuscript.

Referee 2:

We thank the referee for the encouraging assessment and comments.

As requested, we toned down the claims of reactant diffusion as the major contribution to photocatalytic enhancement in different places throughout the main text:

In the abstract, it now reads: **“The introduction of mesopores and the associated formation of new active sites have improved the HER rates”**

In the photocatalytic section (page 8), we changed the paragraph to read:

“The rates of the heated samples (5 hours) are considerably higher than those of the as-prepared samples. Note that there is a major difference in the extent of rate enhancement that seems to be linked to the type of porosity. The samples with cavity-type pores (2%-10%NH₂-MIL) exhibit a rate increase of about 1.4-1.5x, while those containing interconnected fracture-type pores (50% and 80%NH₂-MIL) experience a much more pronounced 4-5 times enhancement. Changes in size and location of the Pt co-catalyst between the samples appear too small (Supplementary Fig. 19) to impact the HER rate to this extent. A contribution by TiO₂ nanoclusters smaller than the detection limit (<1 nm) seems unlikely in light of the complementary characterization methods, but cannot be fully excluded and requires future investigations.

The apparent correlation between rate enhancement and type of mesopores suggests that the formation of additional active sites upon selective ligand removal (i.e., terminal OH groups in the SBU) and their enhanced accessibility via reactant diffusion through large, preferably 3D-interconnected mesopores are key to the enhanced performance. This hypothesis is supported by the increased formation of CO₂ in the heat-treated samples (Supplementary Fig. 23), where the sacrificial agent methanol can more easily enter the pores, adsorb and get oxidized at the newly created OH-groups. It further complies with the observed decrease in rate with samples treated for 20 hours, which is explained by surface corrosion leading to a partial pore collapse and blockade of active sites (Supplementary Fig. 20). In addition, a more efficient charge separation at the newly formed active sites (hole trapping via methoxy formation) as well as changes in electronic structure of the MOFs, as revealed by diffuse reflectance spectroscopy (DRS) (Supplementary Fig. 25b), will likely affect the photocatalytic performance as well. Although these observations seem quite conclusive, future studies are required to investigate the effects of selective ligand removal on the band positions and defect structure, the nature of active Lewis sites⁵³ and the charge separation dynamics in order to fully understand their role in enhancing the photocatalytic activity in heat-treated MOFs.”

In the conclusion section (page 10), we revised to read:

“We attribute this mainly to the formation of new active sites, their enhanced accessibility via reactant diffusion through the interconnected mesopores and an improved charge separation efficiency. It remains for future studies to evaluate the impact of selective ligand removal on changes in band positions and inter-band gap states as well as on the nature of Lewis sites and their role as adsorption sites and charge recombination centres.”